# Vitamin D Supplementation: Practical Advice in Different Clinical Settings

**DOI:** 10.3390/nu17050783

**Published:** 2025-02-24

**Authors:** Giulia Bendotti, Emilia Biamonte, Paola Leporati, Umberto Goglia, Rosaria Maddalena Ruggeri, Marco Gallo

**Affiliations:** 1Endocrinology and Metabolic Diseases Unit, SS. Antonio e Biagio e Cesare Arrigo Teaching Hospital, 15121 Alessandria, Italy; emilia.biamonte@ospedale.al.it (E.B.); paola.leporati@ospedale.al.it (P.L.); marco.gallo@ospedale.al.it (M.G.); 2Endocrinology and Diabetology Unit, Local Health Authority CN1, 12100 Cuneo, Italy; umberto.goglia@gmail.com; 3Endocrinology Unit, Department of Human Pathology of Adulthood and Childhood DETEV, University of Messina, 98121 Messina, Italy; rmruggeri@unime.it

**Keywords:** vitamin D supplementation, chronic kidney disease, pregnancy, obesity, liver disease, intestinal malabsorption

## Abstract

A global deficiency in vitamin D is now widely prevalent. Extensive scientific research has provided compelling evidence of the detrimental effects of vitamin D deficiency on the skeletal system. Furthermore, vitamin D supplementation not only helps prevent bone fractures but may also slow the progression of various conditions, such as diabetes, autoimmune disorders, and cardiovascular disease. Achieving optimal circulating vitamin D levels can be challenging, particularly in certain clinical scenarios. Moreover, the effect of vitamin D supplementation varies depending on factors such as body weight, pregnancy status, absorption capacity, metabolic rate, and renal function. This review aims to explore which vitamin D formulations are most effective in specific clinical contexts where reaching adequate vitamin D levels may not be straightforward.

## 1. Introduction

Vitamin D exists primarily in two forms: vitamin D3 (cholecalciferol), synthesized in the skin upon exposure to ultraviolet B rays, and vitamin D2 (ergocalciferol), derived from plant sources. Both forms are metabolized in the liver to produce 25-hydroxyvitamin D [25(OH)D], the most reliable marker of vitamin D status in the body [1,2]. From there, 25(OH)D is further converted in the kidneys to its biologically active form, 1,25-dihydroxyvitamin D [1,25(OH)2D], which exerts its effects through the vitamin D receptor expressed in various tissues [2,3] (Figure 1). One of the critical functions of vitamin D is to enhance intestinal calcium and phosphorus absorption, ensuring adequate calcium levels for bone mineralization. Vitamin D, traditionally recognized for its pivotal role in calcium homeostasis and skeletal health, has garnered increasing attention due to its wide-ranging effects on human physiology, often referred to as pleiotropic actions. It is well known that a deficiency in vitamin D is closely linked to several bone diseases, including rickets in children and osteoporosis in adults [3,4]. However, beyond its skeletal actions, recent studies have expanded the understanding of vitamin D’s role in extraskeletal functions, such as immune modulation, regulation of cell proliferation, and metabolic and cardiovascular health [3,5]. Given the high prevalence of vitamin D deficiency worldwide, especially in regions with limited sunlight exposure, supplementation is commonly recommended to maintain optimal levels. Plasma 25-hydroxyvitamin D [25(OH)D] concentrations below 20 ng/mL are generally considered deficient, while concentrations between 30 and 60 ng/mL are deemed optimal for most populations [1,5]. Various forms of vitamin D are available for supplementation, including cholecalciferol (D3), calcifediol [25(OH)D], and calcitriol [1,25(OH)2D] [1] (Table 1).

The recommended daily intake of vitamin D and the choice of one form over another varies based on individual factors, including age, health status, and geographic location, with the goal of achieving or maintaining adequate plasma concentrations of 25(OH)D.

## 2. Vitamin D and Obesity

The relationship between vitamin D and obesity is very complex regarding its role in specific clinical features, depending on ethnicity, age, body mass index (BMI), total fat mass, and subcutaneous and visceral adiposity [6]. Recent evidence demonstrated that in obese subjects, free plasma 25(OH)D and 1,25(OH)2D concentrations are lower than in normal-weight individuals [7], with a mechanism related to a reduced release of 1,25(OH)2D from subcutaneous adipose tissue, via β-adrenergic stimulation [8]. Vitamin D Is a lipophilic molecule, and it is stored in lipid droplets [9]. This storage mechanism leads to a reduced bioavailability of vitamin D in circulation and a delayed release of vitamin D into the bloodstream, impacting its effectiveness [9]. However, in obesity, there are conflicting reports about whether the enzymes responsible for vitamin D metabolism (CYP27B1, CYP24A1, CYP2R1, and CYP2J2) are upregulated or downregulated, potentially altering the metabolism and degradation of vitamin D within fat stores [9]. Moreover, another key factor affecting vitamin D concentrations in obese individuals is its concentration in the bloodstream. This means that even with adequate vitamin D intake, serum 25(OH)D concentrations may be misleadingly low. Due to a greater storage capacity in visceral adipose tissue and due to a slower release from adipose tissue, an increased requirement for vitamin D supplementation is necessary to maintain normal serum levels [9]. Also, a diet low in fish and meat and high in carbohydrates could contribute to the low levels of vitamin D found in obese people [10].

Cholecalciferol requires hepatic 25-hydroxylation for conversion to 25(OH)D, a process that may be impaired in obese individuals due to hepatic dysfunction or metabolic irregularities [9,10]. Calcifediol bypasses this step and can be used regardless of BMI and vitamin D baseline status. Thanks to better intestinal absorption and a greater hydrophilic feature, calcifediol results in a faster and more predictable increase in plasma 25(OH)D concentrations, making it particularly advantageous for patients with obesity [11]. Calcifediol can be used for supplemental treatment at doses of 5 mcg/day in children 3–10 years and doses of up to 10 mcg per day for children ≥11 years and adults (50 mcg/per day) [12]. Some studies showed an association between low plasma 25(OH)D concentrations and a higher prevalence of obesity in different ages [13]. Systematic review and meta-analysis of controlled trials demonstrated that plasma 25(OH)D concentrations are raised by weight loss, even without vitamin D supplementation [14]. In human interventional studies, the interpretation of available evidence is not coherent due to differences in baseline plasma 25(OH)D concentrations at baseline, supplementation duration, and vitamin D dose across populations. A meta-analysis by Pathak and colleagues evaluated 12 trials and concluded that vitamin D supplementation did not significantly influence body weight, fat mass, percentage of fat mass, and lean body mass [15]. In line with these results, evidence showed that cholecalciferol intake had no significant impact on the percentage of fat mass [16]. According to Barbara J Boucher, the primary challenge in evaluating randomized controlled trials about vitamin D supplementation is considering vitamin D a drug rather than a nutrient [17]. Another important issue is that, in most vitamin D randomized controlled trials, participants are enrolled with high baseline 25(OH)D concentrations, receive low vitamin D doses, and outcomes are analyzed based on intention-to-treat rather than the achieved 25(OH)D concentration [18]. Furthermore, there are other dietary factors, such as magnesium (essential for enzymatic function), vitamin A (important in vitamin D molecular signaling), and vitamin K (a modulator of vitamin D efficacy), that are often overlooked in traditional randomized controlled trials [17]. An interesting analysis of the effects of supplementation of cholecalciferol (2000 IU/day) was performed in a subset of the Vitamin D and Omega-3 Trial (VITAL). Prior to randomization, an inverse relationship between serum total plasma 25(OH)D concentrations and BMI was observed [19]. After two years, increases in plasma 25(OH)D concentrations were very similar for all three BMI groups [19]. Finally, a systematic review evaluated the effect of vitamin D supplementation on the lipid profile in subjects overweight and obese, showing that vitamin D has no effect on the LDL-c, HDL-c, and total cholesterol concentration, but a significant effect on hypertriglyceridemia was observed [20]. In conclusion, calcifediol supplementation offers distinct advantages for managing vitamin D deficiency in obese individuals. Its rapid efficacy, reduced dependency on liver metabolism, and resistance to adipose sequestration offer a superior alternative to cholecalciferol. While 25(OH)D remains the standard marker for assessing vitamin D status, its interpretation in obese individuals should be carried out with caution. Other indicators, such as PTH levels, may provide additional insights into vitamin D sufficiency. In human interventional studies, evidence of a direct role of vitamin D supplementation in BMI, fat mass, and percentage of fat mass is inconsistent, whereas, in the literature, some evidence appears around an improvement of markers of cardiovascular risk factors [9]. For clinical practice, selecting calcifediol can ensure better outcomes in terms of achieving optimal plasma 25(OH)D concentrations and mitigating the health risks associated with deficiency.

## 3. Vitamin D and Intestinal Malabsorption

Many gastrointestinal conditions, as well as several surgical procedures (such as bariatric surgery), can lead to impaired gastrointestinal absorption of 25(OH)D, sometimes associated with reduced calcium absorption and potentially leading to secondary hypoparathyroidism and negative skeletal outcomes [21,22] (Table 2).

Celiac disease (CD) is the most prevalent malabsorptive disease (global prevalence: ~1%). Villus atrophy and lymphocytic gut mucosal infiltration associated with long-standing CD impair calcium and vitamin D absorption, frequently leading to low bone mineral density (BMD), which in turn may lead to secondary osteoporosis and increased fracture risk [23]. Conversely, a strict gluten-free diet, associated with calcium and vitamin D supplementation, improves BMD, controls secondary hyperparathyroidism, and improves skeletal outcomes. Accordingly, plasma 25(OH)D concentrations should be monitored in every patient with CD, and supplementation should be provided to restore and maintain normal levels (>30 ng/mL). To this end, pharmacological doses of vitamin D (from 600 to 4000 IU/day) should be provided, according to the activity of the disease and vitamin D plasma concentrations. Interestingly, according to Trasciatti et al., in a preclinical model, the administration of cholecalciferol was able to significantly reduce the presence of intestinal mucosal lesions and increase villi length in a dose-dependent manner, suggesting a direct ameliorative role of vitamin D supplementation on CD [24].

Hypovitaminosis D is a common finding in patients with inflammatory bowel diseases (IBDs, i.e., Crohn’s disease and ulcerative colitis), with several mechanisms potentially contributing. These include dietary restriction, impaired absorption of nutrients and bile salts, chronic low-grade inflammation, and glucocorticoid treatment. These findings are deemed responsible for a higher risk of bone fragility, lower BMD levels, worse bone quality, and increased fracture risk [25]. Conversely, vitamin D supplementation has been shown to prevent bone loss in patients with IBD. As well as, for patients with Crohn’s disease, normal circulating plasma 25(OH)D concentrations should be guaranteed through therapeutic supplementation (>800 IU/day), if needed [26,27,28].

Due to the increasing prevalence of obesity and metabolic diseases, which have reached pandemic proportions, bariatric surgical procedures have progressively become more and more common in the Western world. People undergoing bariatric surgery are commonly characterized by low pre-surgical vitamin D levels. Moreover, restrictive surgical procedures (such as laparoscopic sleeve gastrectomy) and bariatric procedures with a malabsorptive component (such as biliopancreatic diversion and at lower degrees— Roux-en Y gastric bypass) are almost always characterized by variable degrees of malabsorption, with the last ones clearly associated with the highest risk of fracture [26]. To prevent bone loss and maintain skeletal mass, it is fundamental to maintain adequate postoperative circulating levels of vitamin D, even if no guidelines are specifically available for vitamin D supplementation in this setting. It is noteworthy that significant postoperative BMD reductions are commonly found despite even high-dose (>2000 IU daily) oral vitamin D supplementations, thus suggesting the potential role of intramuscular vitamin D administration [21,26,27,28].

In addition to the well-recognized effects of gastrointestinal malabsorption on plasma 25(OH)D concentrations, a potential bidirectional relationship between vitamin D deficiency and malabsorptive conditions has been suggested. Indeed, low plasma 25(OH)D concentrations have been proposed as a biomarker for disease activity and as a predictor of poor clinical outcomes in IBDs. Conversely, subjects with higher plasma 25(OH)D concentrations were significantly more likely to show endoscopic remission than those with lower levels. For instance, vitamin D receptors, highly expressed in gut cells, may mediate the immunomodulatory actions of vitamin D on the gut mucosa [22]. Furthermore, vitamin D is increasingly accomplished in a direct role impacting both innate and adaptive immune responses. Vitamin D has been shown to enhance the production of anti-inflammatory cytokines, such as IL-4 and IL-10, for instance [29]. Beyond exerting its biological effects on the intestine by maintaining mucosal barrier integrity and modulating the immune system, vitamin D also affects the composition of fecal microbiota, with high plasma 25(OH)D concentrations generally associated with higher levels of beneficial bacteria and reduced levels of pathogenic bacteria [30,31]. Finally, vitamin D deficiency may unfavorably affect response to biological therapy in IBDs, with an increased risk of primary non-response, as well as secondary failure to these drugs [22].

Furthermore, vitamin D supplementation may play a role in maintaining weight loss and its beneficial metabolic consequences after bariatric surgery due to the hypothesized anti-inflammatory actions of vitamin D on adipose tissue and the positive effects on muscle structure and body composition [22].

In conclusion, regular monitoring and appropriate supplementation not only improve bone health in gastrointestinal malabsorption syndromes but may also enhance disease outcomes through immunomodulatory effects and microbiota modulation.

## 4. Vitamin D and Liver Disease

Vitamin D insufficiency and deficiency are prevalent in chronic liver diseases, correlating with the severity of liver failure as measured by Child-Pugh and Mayo End stage Liver Disease (MELD) scores [32]. Several factors contribute to this issue, including malnutrition, reduced sun exposure, malabsorption, decreased synthesis of binding proteins (i.e., VDBP) and albumin, and the insufficient capability of hepatocytes to hydroxylate vitamin D. Specifically, the causes of liver disease and its treatments can influence various aspects of vitamin D metabolism [21]. Impaired intestinal absorption is the primary cause of hypovitaminosis D in patients with cholestatic liver disease, such as primary biliary cirrhosis and primary sclerosing cholangitis, due to poor absorption of fat-soluble vitamin D and disrupted enterohepatic circulation. Additionally, using cholestyramine to treat pruritus and hypercholesterolemia in these patients can further negatively impact the intestinal absorption of vitamin D [33]. Some evidence suggests that low plasma 25(OH)D concentrations may correlate with clinical disease severity in patients with parenchymal liver damage, such as alcoholic liver disease, MAFLD (metabolic dysfunction-associated fatty liver disease), and chronic hepatitis C. Concordantly, in MAFLD, autoimmune and chronic viral hepatitis histological findings of hepatic steatosis, hepatic necroinflammation, and fibrosis are significantly more pronounced in patients with lower vitamin D [34]. Moreover, patients with severe liver disease resulting in at least 80% functional impairment, such as cirrhosis, may have compromised hepatic metabolism of vitamin D, leading to reduced efficiency in 25-hydroxylation or increased catabolism of calcifediol [33,35].

Low plasma 25(OH)D concentrations can lead to secondary hyperparathyroidism and are associated with hepatic osteodystrophy and subsequent osteoporosis, increased bone turnover, and a high risk of fractures [36]. Instead, osteomalacia is rarely observed in clinical hepatology practice, except in advanced hepatic disease with severe malabsorption [37]. While the link between hypovitaminosis D and liver disease has emerged in recent years and mechanisms are being clarified, there remains a lack of specific guidelines and target reference values. In clinical practice, vitamin D levels should be periodically monitored in patients with liver disease, and if plasma 25(OH)D concentrations are below 30 ng/mL, it is advisable to consider supplementation. Generally, administering cholecalciferol at an average dose of 2.000 IU/day is effective in elevating plasma 25(OH)D concentrations across various chronic liver diseases. However, the available evidence is still insufficient to demonstrate effects on liver-related morbidity or mortality [38]. Theoretically, calcifediol can be considered an alternative approach, although evidence of its efficacy remains limited [37]. Indeed, comparative trials between calcifediol and cholecalciferol in this clinical context are lacking. Supplementation is essential in the case of osteoporosis. For severe liver diseases, the recommendation for using cholecalciferol or, alternatively, calcifediol at similar dosages as in osteoporotic patients has been suggested [21]. Some leak suggestions may indicate vitamin D supplementation’s potential role in enhancing immune responses to antiviral therapy in HCV [39] and preventing acute rejection post-liver transplant [40].

In summary, it is reasonable to consider using calcifediol, which is already 25-hydroxylated, in patients with severe liver disease. However, surprisingly, there is limited literature on the use of calcifediol. Specifically, no trials have compared calcifediol and cholecalciferol in patients with severe liver disease. For most patients, cholecalciferol should be sufficient, but in severe cases, calcifediol may be helpful. While standard guidelines are lacking, maintaining plasma 25(OH)D concentrations above 30 ng/mL through supplementation, typically with cholecalciferol, is widely recommended.

## 5. Vitamin D and Chronic Kidney Disease

In chronic kidney disease (CDK) patients, especially those in pre-dialysis stages, vitamin D deficiency is primarily driven by reduced renal function, limiting the kidney’s ability to activate 25(OH)D into its active form, 1,25-dihydroxyvitamin D (calcitriol). As the disease progresses, phosphate retention and reduced calcium absorption exacerbate this deficiency, leading to an overproduction of parathyroid hormone (PTH), which further disrupts mineral metabolism, leading to conditions such as CKD-mineral and bone disorder [41,42].

Kidney Disease Global Outcome (KDIGO) guidelines recommend routine monitoring and supplementation of vitamin D in CKD patients. However, there remains a lack of consensus on optimal plasma 25(OH)D concentrations, and different guidelines suggest varying thresholds [41,43].

For CKD patients, especially those in stages 3–4, the Kidney Disease Outcomes Quality Initiative (KDOQI) recommends using cholecalciferol to correct vitamin D insufficiency or deficiency. The dosage is typically adjusted based on serum levels of 25(OH)D (Table 3). If plasma 25(OH)D concentrations are <5 ng/mL, the recommended dose is 50.000 IU of cholecalciferol orally every week for 12 weeks, followed by a monthly dose. If plasma 25(OH)D concentrations are 5–15 ng/mL, the recommended dose is 50.000 IU orally every week for 4 weeks, then monthly. Instead, if plasma 25(OH)D concentrations are 16–30 ng/mL, they suggest 50.000 IU orally once a month [43]. Studies suggest that cholecalciferol supplementation is effective in raising serum plasma 25(OH)D concentrations and in preventing secondary hyperparathyroidism in non-dialysis CKD patients, particularly in the early stages (CKD stage 2–3) [44,45,46]. When cholecalciferol is not sufficient to address severe PTH elevation or fully restore 1,25(OH)2D levels, calcitriol or calcifediol are often employed to directly manage PTH [47]. Dialysis patients frequently exhibit low levels of both 25(OH)D and 1,25(OH)2D, which are associated with high levels of PTH [48]. Studies highlight the need for combination therapies in dialysis patients, where both cholecalciferol and active vitamin D analogs (such as calcitriol) are used [48]. In this population, cholecalciferol is preferred, with recommendations ranging from 25.000 to 50.000 IU every two weeks or 100.000 IU monthly to maintain plasma 25(OH)D concentrations above 30 ng/mL [41,43]. Studies have shown that supplementation with high doses of cholecalciferol (e.g., 50.000 IU twice weekly) can help reduce serum parathyroid hormone (PTH) levels and improve cardiovascular markers [41]. However, in a study comparing cholecalciferol and calcitriol, cholecalciferol was shown to effectively raise plasma 25(OH)D concentrations but was less effective than calcitriol in lowering PTH [45]. Indeed, calcitriol therapy has been shown to significantly reduce PTH levels by up to 50% in dialysis patients [41]. Despite this recommendation, recent studies underscore the importance of frequent, daily administration for achieving optimal serum levels and potential extraskeletal benefits. Although the use of bolus dosing is convenient, it may result in higher activation of the 24-hydroxylase enzyme, accelerating vitamin D degradation and potentially limiting its therapeutic effects [1]. Another study indicates that maintaining sufficient magnesium levels is important when addressing vitamin D deficiency in patients with renal insufficiency [49].

In conclusion, while native vitamin D (cholecalciferol) is recommended for CKD patients to prevent secondary hyperparathyroidism and maintain bone health (Figure 2), dialysis patients often require higher doses and the use of active vitamin D analogs to achieve optimal outcomes. Supplementing cholecalciferol or calcifediol allows the body to maintain a regulated vitamin D pool, while direct calcitriol supplementation can lead to excessive calcium levels and poor long-term management.

Thus, while calcitriol might seem like the logical choice, cholecalciferol or calcifediol are preferred in CDK patients (stages 2–4) because they offer better regulation, longer duration of action, and a lower risk of complications.

Tailoring the choice of vitamin D supplementation based on CKD stage, PTH levels, and calcium–phosphate balance is essential for optimal patient outcomes.

## 6. Vitamin D and Pregnancy/Breastfeeding

In recent years, growing evidence suggests that adequate vitamin D levels during pregnancy and lactation are essential both for maternal health and for fetal development and infant outcomes.

During pregnancy, vitamin D plays a crucial role in maternal bone health and the regulation of calcium levels, which are needed for fetal skeletal development [50]. Studies suggest that vitamin D is important for immune function by reducing the risk of infections and autoimmune diseases during pregnancy [51]. Vitamin D deficiency in pregnancy is associated with adverse outcomes, including preeclampsia, gestational diabetes, preterm birth, and impaired fetal growth [52,53,54]. Therefore, adequate plasma 25(OH)D concentrations are recommended for the prevention of these conditions [54]. For breastfeeding mothers, insufficient vitamin D can lead to low levels in breast milk; moreover, vitamin D content in human breast milk is clearly low and insufficient to obtain the recommended intake of 400 IU daily, thus increasing the risk of rickets and infections in infants [55].

Therapeutic strategies aim to optimize maternal and neonatal vitamin D status by targeting plasma 25(OH)D concentrations. The recommended vitamin D intake during pregnancy varies by country and health organization. According to previous endocrine society guidelines, plasma vitamin 25(OH) concentrations were considered sufficient when >30–40 ng/m, while the current 2024 guidelines did not find adequate evidence to confirm these thresholds, both for pregnant women and the general population [56,57]. In contrast, the IOM recommended that concentrations of at least 20 ng/mL should be considered physiologically adequate. Therefore, deficiency is defined as serum 25(OH)D concentrations <20 ng/mL, insufficient concentrations range between 21 and 29 ng/mL, and sufficiency is considered as >30 ng/mL [58,59]. Maintaining optimal plasma 25(OH)D concentrations is particularly important in the first trimester, as deficiency during this period has been linked to poor implantation and an increased risk of miscarriage [60].

As far as concern the therapy, the Institute of Medicine (IOM) recommends a daily intake of 600 IU (15 µg) of vitamin D for pregnant women [58,59], while other sources, including the Endocrine Society, suggest that higher doses (4000 IU/day) may be necessary to achieve sufficient serum concentrations (≥30–40 ng/mL) without adverse effects, particularly for women at risk of deficiency [61]. According to the World Health Organization (WHO), pregnant women should be encouraged to receive adequate nutrition, which is best achieved through the consumption of a healthy, balanced diet [62]. The American College of Obstetricians and Gynecologists (ACOG) and the International Federation of Gynecology and Obstetrics recommend supplementing 250–600 IU/day cholecalciferol during pregnancy as a standard. They also recommend that vitamin D deficiency should be treated with vitamin D3 at doses of 1.000–2.000 IU/day [63,64]. As regards to vitamin D-sufficient population, a large study on 1164 infants assessed at 1 year of age demonstrated that maternal vitamin D supplementation from mid-pregnancy until birth or until 6 months postpartum did not improve fetal or infant growth [65].

During lactation, Vitamin D is transferred to the infant through breast milk, although in relatively small amounts. The vitamin D content of breast milk is directly influenced by the mother’s vitamin D status. In cases of maternal deficiency, breast milk may not provide enough amounts of vitamin D for the infant, leading to an increased risk of rickets and other developmental issues [55]. During breastfeeding, maternal supplementation can enhance vitamin D content in breast milk. Standard doses of 1500–2000 IU/day are often inadequate to meet neonatal needs, whereas high-dose maternal supplementation (6000–6400 IU/day) has been shown to increase milk concentrations to levels that could obviate the need for direct infant supplementation [66]. However, direct supplementation of the infant with 400 IU/day remains the most widely endorsed approach due to its simplicity and efficacy in preventing vitamin D deficiency in neonates [55].

Despite the clear benefits, optimal strategies for vitamin D supplementation in pregnancy and breastfeeding remain controversial, with ongoing debates on variability in optimal dosing, adherence, and the lack of a universal consensus on supplementation protocols.

## 7. Conclusions

Vitamin D plays a critical role in various physiological processes, extending beyond its well-known effects on skeletal health. Its importance in maintaining bone mineralization, modulating immune responses, supporting cardiovascular health, and improving metabolic outcomes highlights its systemic relevance. However, vitamin D deficiency remains a global health concern, influenced by factors such as obesity, gastrointestinal malabsorption, chronic liver and kidney diseases, and life stages like pregnancy and lactation.

Recent findings underline the complexity of vitamin D metabolism and the varying efficacy of supplementation strategies across different clinical settings. In obesity and malabsorptive conditions, calcifediol emerges as a more effective alternative to cholecalciferol due to its superior bioavailability and greater hydrophilic feature. In hepatic and renal chronic diseases, supplementation with cholecalciferol is preferred and should be tailored to disease severity and baseline plasma 25(OH)D concentrations. The same strategy can be applied for pregnant and breastfeeding women, maintaining adequate plasma 25(OH)D concentrations for maternal and fetal health.

In conclusion, clinicians should consider factors such as baseline plasma 25(OH)D concentrations, geographical location, and dietary habits to develop personalized strategies for addressing vitamin D deficiency, ensuring effective prevention and treatment tailored to individual needs.

Further research and international consensus are necessary to refine guidelines and ensure effective prevention and management of vitamin D deficiency during these critical life stages.

## Figures and Tables

**Figure 1 nutrients-17-00783-f001:**
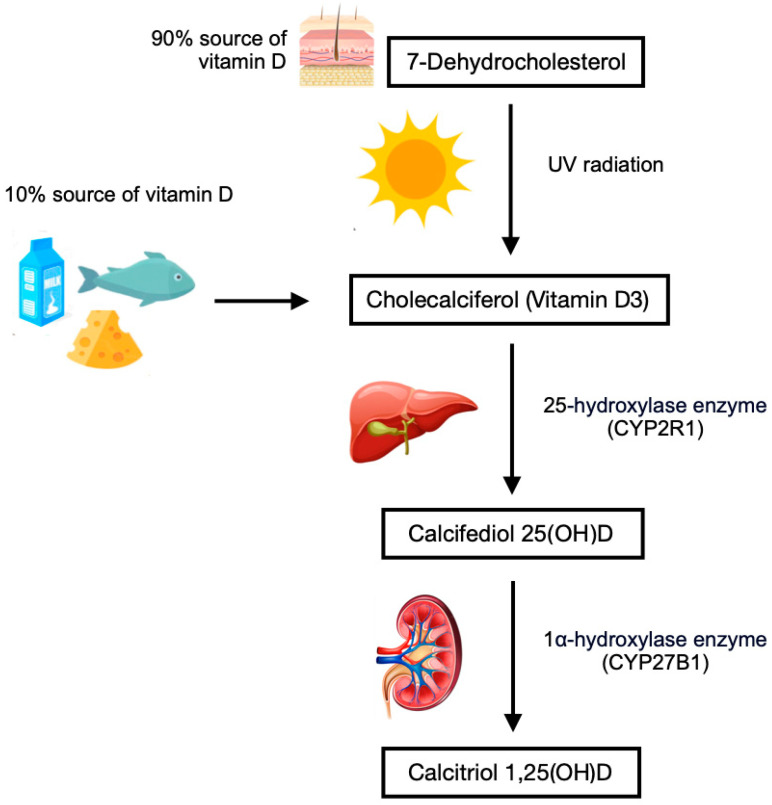
Vitamin D synthesis.

**Figure 2 nutrients-17-00783-f002:**
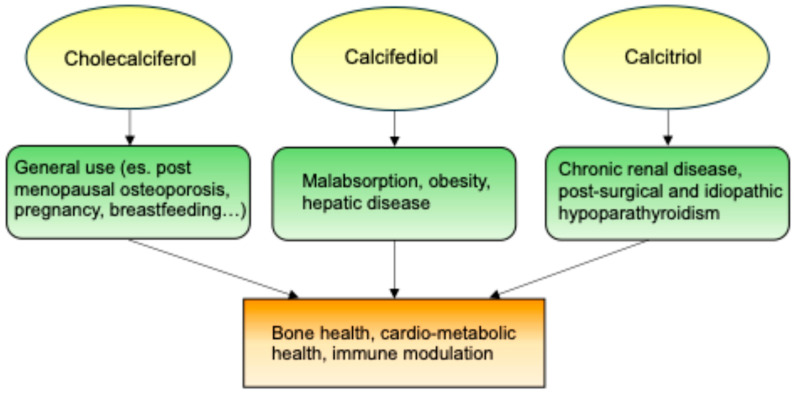
Clinical applications of vitamin D supplementation.

**Table 1 nutrients-17-00783-t001:** Main characteristics of the principal forms of vitamin D.

	Cholecalciferol (D3)	Calcifediol 25(OH)D3	Calcitriol 1,25(OH)₂D3
Definition	Inactive form of vitamin D3	Intermediate metabolite of vitamin D3	Active form of vitamin D3
Synthesis	Synthesized in the skin upon UVB exposure	Produced by conversion of cholecalciferol in the liver	Produced by conversion of calcifediol in the kidneys
Site of Production	Skin	Liver	Kidney
Main Function	Precursor to active vitamin D	Indicator of vitamin D status in the body	Regulates calcium and phosphorus levels in the blood
Biological Activity	Inactive	Inactive	Active
Duration of Action	Long (weeks to months)	Long (2–4 weeks)	Short (4–6 h)
Clinical Use	Supplementation for vitamin D deficiency	Assessment of vitamin D status, supplementation for vitamin D deficiency	Treatment of hypocalcemia (CDK, post-surgical and idiopathic hypoparathyroidism)
Conversion	Requires conversion to calcifediol, then to calcitriol	Converted into calcitriol	Final form, no further conversion needed
Regulation	Depends on sun exposure and dietary intake	Regulated by availability of cholecalciferol	Regulated by PTH, calcium, and phosphate levels
Administration	Oral (capsules, drops, or sublingual film) or intramuscular	Oral (capsules or drops)	Oral or intravenous

**Table 2 nutrients-17-00783-t002:** Main causes of gut malabsorption.

Causes of Impaired Gastrointestinal Absorption of 25(OH)D	Gastrointestinal Conditions
Decreased absorptive intestinal surface	Celiac disease; tropical sprue;
IBD	Crohn’s disease; ulcerative colitis
Bariatric surgery	Sleeve gastrectomy; Roux-en Y gastric bypass; biliopancreatic diversion;
Infectious causes of chronic diarrhea	e.g., Whipple’s disease; HIV; parasitic infections; enteropathogenic bacteria and viruses;
Other causes of small bowel disease or resection	Short bowel syndrome; radiation enteropathy; etc.
Gastrointestinal malignancies	e.g., Kaposi sarcoma; gut lymphoma; etc.
Small intestinal dysbiosis	Small bowel bacterial overgrowth
Pancreatic exocrine insufficiency	Chronic pancreatitis; pancreatectomy; cystic fibrosis and other hereditary disorders (e.g., Schwachman syndrome);
Gastric hypersecretory status	e.g., Zollinger–Ellison syndrome;
Neuroendocrine neoplasms [Gallo 2019]	e.g., Carcinoid syndrome; VIPoma (Verner–Morrison syndrome); glucagonoma; somatostatinoma; medullary thyroid carcinoma;
Drugs [Kupisz-Urbańska 2021]	Laxatives; lipase inhibitors (e.g., orlistat; cetilistat); SSRIs; NSAIDs; antibiotics; somatostatin analogs, systemic chemotherapy (e.g., streptozocin, platinum salts, 5-fluorouracil, etc.), tyrosine kinase inhibitors (e.g., sunitinib, vandetanib, etc.); magnesium-containing products; etc.
Disorders of bile acid metabolism	e.g., Advanced cirrhosis; cholestasis;
Diet	Artificial sweeteners; caffeine; etc.

**Table 3 nutrients-17-00783-t003:** Suggested vitamin D supplementation based on baseline plasma 25(OH)D concentrations.

Plasma 25(OH)D Concentrations			
	Significant Deficiency: <12 ng/mL (0–30 nmol/L)	Moderate Vitamin D Deficiency: 13–29 ng/mL (31–72 nmol/L)	Adequate Levels: ≥30 ng/mL (75 nmol/L)
Suggested cholecalciferol dose	A total of 300,000 IU in cumulative dose to administer over a period of 12 weeks. The dose can be divided into daily (~3500 IU per day), weekly (25,000 IU per week), or monthly doses (50,000 IU every 2 weeks). It is recommended not to exceed boluses of 100,000 IU for safety reasons.	2000 IU per day (i.e., sublingual film, tablets) or ~8 drops per day (considering 250 IU per drop)	Continue 1000 IU daily (i.e., sublingual film, tablets) or ~4 drops per day (considering 250 IU per drop)
Suggested calcifediol dose	1 tablet (0.266 mg) twice a month or calcifediol ~32 drops per week	1 tablet (0.266 mg) once a month or calcifediol ~16 drops per week	Continue calcifediol 0.266 mg once a month or calcifediol ~16 drops per week

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
