# Peer review of "Vitamin D Supplementation: Practical Advice in Different Clinical Settings"

_nutrients, 2025, doi:10.3390/nu17050783_

Round 1
Reviewer 1 Report
Comments and Suggestions for Authors
This article entitled "Vitamin D supplementation: practical advice in different clinical settings", manuscript ID: nutrients-3482004, exhibits the different forms of vitamin D and their beneficial effects in the progression of various disorders. Overall, the article is well-structured with pertinent references and fluent language, but there are major revisions:
- Abstract: It is quite clear and comprehensive but does not concord with the whole manuscript as the keywords can reflect. So, It has to be rewritten to clearly fit with the main manuscript. Check also some expressions used, for example, use "pregnancy status" instead of "pregnant state".
- Introduction: The different forms of Vitamin D were well-stated, as well as their recommended levels to maintain the organism's balance. However, there is no introduction part with the various disorders. the authors should had some transition between the introduction and the other ideas to make the reading experience smooth. In the text, the authors mentioned Ergocalciferol (D2) when citing Table 1, but it's not in the table, why? especially when the authors said it is one of the main forms of Vitamin D.
- Vitamin D and chronic kidney diseases: What does it mean KDIGO? Give the meaning of abbreviations when they are used for the first time.
Comments on the Quality of English Language
The document is well-structured, scientifically sound, and provides a comprehensive discussion on Vitamin D supplementation in various clinical settings. However, there are areas where clarity, grammar, and sentence structure can be improved to enhance readability and professional polish. Below are specific observations and suggestions for improvement:
- Some sentences are too long and contain multiple ideas, making them difficult to read. Breaking them into shorter sentences will improve clarity.
- Rephrasing some sentences to make them more precise. Example: “Besides, the effect of vitamin D supplementation varies based on factors such as body weight, pregnant state, absorption capacity, metabolic differences and renal function.” Better phrasing: “Moreover, the effect of vitamin D supplementation varies depending on factors such as body weight, pregnancy status, absorption capacity, metabolic rate, and renal function.”
- Article usage (e.g., missing or unnecessary "the" or "a") needs attention.
- Missing transitions: Adding linking words like "therefore," "furthermore," "consequently," or "notably" will help improve logical flow.
Author Response
This article entitled "Vitamin D supplementation: practical advice in different clinical settings", manuscript ID: nutrients-3482004, exhibits the different forms of vitamin D and their beneficial effects in the progression of various disorders. Overall, the article is well-structured with pertinent references and fluent language, but there are major revisions:
- Abstract: It is quite clear and comprehensive but does not concord with the whole manuscript as the keywords can reflect. So, It has to be rewritten to clearly fit with the main manuscript. Check also some expressions used, for example, use "pregnancy status" instead of "pregnant state". Thank you for your compliments. We modified the abstract for a better fitting with the whole manuscript. We checked the text according to your suggestions
- Introduction: The different forms of Vitamin D were well-stated, as well as their recommended levels to maintain the organism's balance. However, there is no introduction part with the various disorders. the authors should had some transition between the introduction and the other ideas to make the reading experience smooth. In the text, the authors mentioned Ergocalciferol (D2) when citing Table 1, but it's not in the table, why? especially when the authors said it is one of the main forms of Vitamin D. Thank you for your comment, we decided to eliminate ergocalciferol from the test to focus on the other forms most used in clinical practice.
- Vitamin D and chronic kidney diseases: What does it mean KDIGO? Give the meaning of abbreviations when they are used for the first time. The explanation for the acronym KDIGO, a global nonprofit organization producing international guidelines for kidney diseases, has been added to the text (Kidney Disease Global Outcome).
Comments on the Quality of English Language
The document is well-structured, scientifically sound, and provides a comprehensive discussion on Vitamin D supplementation in various clinical settings. However, there are areas where clarity, grammar, and sentence structure can be improved to enhance readability and professional polish. Below are specific observations and suggestions for improvement:
- Some sentences are too long and contain multiple ideas, making them difficult to read. Breaking them into shorter sentences will improve clarity. Thank you, we shortened some long sentences according to your suggestion.
- Rephrasing some sentences to make them more precise. Example: “Besides, the effect of vitamin D supplementation varies based on factors such as body weight, pregnant state, absorption capacity, metabolic differences and renal function.” Better phrasing: “Moreover, the effect of vitamin D supplementation varies depending on factors such as body weight, pregnancy status, absorption capacity, metabolic rate, and renal function.” Done, thank you.
- Article usage (e.g., missing or unnecessary "the" or "a") needs attention.
- Missing transitions: Adding linking words like "therefore," "furthermore," "consequently," or "notably" will help improve logical flow.
Thank you, we added linking words and corrected the article usage.

Reviewer 2 Report
Comments and Suggestions for Authors
Thank you for the opportunity to review this interesting article that seeks to bring together current evidence on vitamin D within specific conditions, including obesity, malabsorptive disorders, liver disease, kidney disease and pregnancy/breastfeed. Overall the article is well written and provides a useful overview of current literature and in some instances national guidelines. Some sections could be clafiried to aid ease of understanding for the reader. My main comments:
Doses of vitamin D are shown IU/die in some sections- I am not familiar with 'die', do you mean IU/day?
Vitamin D and Obesity: You conclude that calcifediol supplementation is the preferred form of vitamin D supplmentation in this group. However, it's not clear if this has been tested in the trials you have described in this section (most seem to have used cholecalciferol). This section perhaps needs a general check for English language to improve clarity. Is it possible to conclude if 25(OH)D is a reliable marker of vitamin D status in obese people?
Malabsorption section: 'are almost always characterized by variable degrees of malabsorption, with the last ones clearly associated with the highest risk of fracture' I think this statement needs a reference.
'Also in obesity, vitamin D supplementation may play a role in maintaining weight
loss and its beneficial metabolic consequences after bariatric surgery, due to the hypothesized anti-inflammatory actions of vitamin D on adipose tissue, and to the positive effects
on muscle structure and body composition' - I would say this relates to bariatric surgery rather than obesity, otherwise it seems to contradict some studies shown in the obesity section.
Liver disease and kidney disese: these sections are very well written and clear. Please describe the abbreviation KDIGO in full.
Pregnancy/breastfeeding: Very well written and useful summary. Can you clarify if the recommended vitamin D levels cited from the Endocrine society are specifically in pregnancy or are these the levels for the general population.
Thank you
Author Response
Thank you for the opportunity to review this interesting article that seeks to bring together current evidence on vitamin D within specific conditions, including obesity, malabsorptive disorders, liver disease, kidney disease and pregnancy/breastfeed. Overall the article is well written and provides a useful overview of current literature and in some instances national guidelines. Some sections could be clafiried to aid ease of understanding for the reader. My main comments:
Doses of vitamin D are shown IU/die in some sections- I am not familiar with 'die', do you mean IU/day?
Yes, “die” has been replaced by “day” throughout the manuscript. Sorry for the inconvenience.
Vitamin D and Obesity: You conclude that calcifediol supplementation is the preferred form of vitamin D supplementation in this group. However, it's not clear if this has been tested in the trials you have described in this section (most seem to have used cholecalciferol). This section perhaps needs a general check for English language to improve clarity. Is it possible to conclude if 25(OH)D is a reliable marker of vitamin D status in obese people?
Thank you for your brilliant observation. We improved the English language, and we added the answer to your question in the text.
Malabsorption section: 'are almost always characterized by variable degrees of malabsorption, with the last ones clearly associated with the highest risk of fracture' I think this statement needs a reference.
Thank you for your comment. We think that the evidence showing that restrictive surgical procedures (especially those with a malabsorptive component) are associated with variable degrees of malabsorption is well-known, and we needed to limit the number of references. However, we added [26] to this sentence.
'Also in obesity, vitamin D supplementation may play a role in maintaining weight
loss and its beneficial metabolic consequences after bariatric surgery, due to the hypothesized anti-inflammatory actions of vitamin D on adipose tissue, and to the positive effects on muscle structure and body composition' - I would say this relates to bariatric surgery rather than obesity, otherwise it seems to contradict some studies shown in the obesity section.
Thank you, of course. We modified the sentence accordingly.
Liver disease and kidney disease: these sections are very well written and clear. Please describe the abbreviation KDIGO in full.
Thanks, we describe the abbreviation it in the text.
Pregnancy/breastfeeding: Very well written and useful summary. Can you clarify if the recommended vitamin D levels cited from the Endocrine society are specifically in pregnancy or are these the levels for the general population.
We thank the reviewer for the suggestion. A sentence was added in the text.

Reviewer 3 Report
Comments and Suggestions for Authors
- While reading the article, I felt that the essence was to show that calcifediol emerges as a more effective alternative to cholecalciferol due to its superior bioavailability and resistance to sequestration in adipose tissue. However, these aspects were not thoroughly explored. It would be appropriate to dedicate a section specifically to explain in detail how sequestration in adipose tissue occurs.
- Your review starts well by presenting the main characteristics of the principal forms of vitamin D. The table was well conducted.
- When reaching the topic "Vitamin D and Obesity," I had the impression that the reader was not properly prepared for it, as obesity is not even mentioned in the title.
- (2000 IU/die)?
- I would like you to explain "resistance to adipose sequestration" in more detail.
- Improve the sentence that says "increasing doses of cholecalciferol improved villi length, morphology, atrophy." Doesn't it sound strange to say "improve atrophy"?
- It is a well-established fact that low 25(OH)D levels have been proposed as a biomarker for disease, but I felt little novelty in the study. What is it proposing that is new?
- In Table 3, there are references for 25(OH)D₃ levels, but are there any for the other isoforms?
- If calcitriol is produced in the kidneys, why is it not the most suitable supplement for people with kidney problems? From the referenced articles (39 and 41), it seems that cholecalciferol is preferred. This seems contradictory. If the kidneys are already malfunctioning, the production of calcitriol could potentially be impaired (the problem is not with cholecalciferol, as it is dependent on UVB exposure as well as skin synthesis).
- Along the same line, it would also make more sense to ingest calcifediol for people with liver problems. It appears that the recommendation is to use cholecalciferol in cases of severe liver diseases, while calcifediol can be considered an alternative approach. Could you elaborate on this?
Author Response
- While reading the article, I felt that the essence was to show that calcifediol emerges as a more effective alternative to cholecalciferol due to its superior bioavailability and resistance to sequestration in adipose tissue. However, these aspects were not thoroughly explored. It would be appropriate to dedicate a section specifically to explain in detail how sequestration in adipose tissue occurs. Thank you for pointing this out. We have implemented the chapter “Vitamin D and obesity” with the changes you requested.
- Your review starts well by presenting the main characteristics of the principal forms of vitamin D. The table was well conducted. Thank you!
- When reaching the topic "Vitamin D and Obesity," I had the impression that the reader was not properly prepared for it, as obesity is not even mentioned in the title. We mentioned it in the abstract “.. Moreover, the effect of vitamin D supplementation varies depending on factors such as body weight…”
- (2000 IU/die)? Die has been replaced by “day”
- I would like you to explain "resistance to adipose sequestration" in more detail. Thank you, we added a better explanation in the text.
- Improve the sentence that says "increasing doses of cholecalciferol improved villi length, morphology, atrophy." Doesn't it sound strange to say "improve atrophy"? According to Trasciatti S et al., in a preclinical model, the administration of cholecalciferol was able to significantly reduce the presence of intestinal mucosal lesions and increase villi length in a dose-dependent manner. However, in our manuscript it is well specified that this refers to an experimental, animal model, and that this effect can be just suggested from this study.
- It is a well-established fact that low 25(OH)D levels have been proposed as a biomarker for disease, but I felt little novelty in the study. What is it proposing that is new? The manuscript is a review of existing evidence, so it adds little to well—established facts
- In Table 3, there are references for 25(OH)D₃ levels, but are there any for the other isoforms? Unfortunately, there aren’t.
- If calcitriol is produced in the kidneys, why is it not the most suitable supplement for people with kidney problems? From the referenced articles (39 and 41), it seems that cholecalciferol is preferred. This seems contradictory. If the kidneys are already malfunctioning, the production of calcitriol could potentially be impaired (the problem is not with cholecalciferol, as it is dependent on UVB exposure as well as skin synthesis). Thank you for this point of view. We have included the explanation in the conclusion of the section regarding chronic kidney disease.
- Along the same line, it would also make more sense to ingest calcifediol for people with liver problems. It appears that the recommendation is to use cholecalciferol in cases of severe liver diseases, while calcifediol can be considered an alternative approach. Could you elaborate on this? Thank you for your brilliant comment. Unfortunately, no trials have compared calcifediol and cholecalciferol in patients with severe liver disease. We add this sentence in the text.

Reviewer 4 Report
Comments and Suggestions for Authors
Furthermore, vitamin D supplementation not only helps prevent bone fractures, but may also slow the progression of various conditions, such as diabetes, autoimmune disorders, cardiovascular disease, and cancer.
Comment: Anything mentioned in the abstract should also be discussed in the text. Cancer is not discussed in the text. I suggest searching Google Scholar for one or two reviews on vitamin D and cancer. Also, one or two reviews regarding cardiovascular disease, diabetes (type 2 diabetes mellitus), and autoimmune diseases.
From there, 25(OH)D is further converted in the kidneys to its biologically active form, 1,25-dihydroxyvitamin D [1,25(OH)2D], which exerts its effects through the vitamin D receptor expressed in various tissues [2,3]
Comment: Please discuss extra-renal conversion to 1,25D. Zehnder and Hewison published an early paper on this topic. It is very important for cancer since the organs with cancer will do it for protection.
One of the critical functions of vitamin D is to enhance intestinal calcium absorption,
Comment: Also phosphorus absorption.
Cholecalciferol
Half-Life in Blood |
Long (weeks to months) |
Comment: I think it is a day or two. What is the reference for this statement.
Calcifediol
Intermediate (2–3 weeks) |
Comment; It is stored in the muscles and put back in the blood when needed, such as in winter:
Mason RS, Rybchyn MS, Abboud M, Brennan-Speranza TC, Fraser DR.The Role of Skeletal Muscle in Maintaining Vitamin D Status in Winter. Curr Dev Nutr. 2019 Jul 25;3(10):nzz087. doi: 10.1093/cdn/nzz087.
Rybchyn MS, Abboud M, Puglisi DA, Gordon-Thomson C, Brennan-Speranza TC, Mason RS, Fraser DR Skeletal Muscle and the Maintenance of Vitamin D Status.Nutrients. 2020 Oct 26;12(11):3270. doi: 10.3390/nu12113270.
Thanks to a better intestinal absorption and a greater hydrophilic feature, calcifediol results in a faster and more predictable increase in vitamin D levels,
Comment: If vitamin D levels were written as 25(OH)D concentration, it would be readily apparent that calcifediol is 25(OH)D, so does not go through the D to 25(OH)D conversion step.
Since “plasma 25(OH)D concentrations” is used a few lines later, please use that instead of vitamin D levels.
According to Barbara J Boucher, the primary challenge in evaluating randomized controlled trials about vitamin D supplementation is considering vitamin D a drug rather than a nutrient [15].
Comment: This statement should be expanded. The problem is that in most vitamin D RCTs, participants are enrolled with high 25(OH)D concentrations, are given low vitamin D doses, and the outcomes are based on intention to treat, not achieved 25(OH)D concentration.
A couple additional references to consider:
Critical Appraisal of Large Vitamin D Randomized Controlled Trials.
Pilz S, Trummer C, Theiler-Schwetz V, Grübler MR, Verheyen ND, Odler B, Karras SN, Zittermann A, März W. Nutrients. 2022 Jan 12;14(2):303. doi: 10.3390/nu14020303.
Intervention Approaches in Studying the Response to Vitamin D3 Supplementation.
Gospodarska E, Ghosh Dastidar R, Carlberg C. Nutrients. 2023 Jul 29;15(15):3382. doi: 10.3390/nu15153382.
Prior to randomization, an inverse relationship between serum total 25(OH)D levels and BMI was observed [16]. After two years, increases in 25(OH)D levels were significantly lower with higher BMI categories [16].
Comment: According to Figure S1(B) in [16], increases in 25(OH)D were very similar for all three BMI groups.
50] C. Palacios, LL. Kostiuk, A. Cuthbert, and J. Weeks, ‘Vitamin D supplementation for women during pregnancy’, Cochrane Database of Systematic Reviews, 2024, doi: 10.1002/14651858.CD008873.pub5
Comment: As this review was not very supportive of vitamin D for pregnant women, I recommend citing this article as well:
Effectiveness of Prenatal Vitamin D Deficiency Screening and Treatment Program: A Stratified Randomized Field Trial.
Rostami M, Tehrani FR, Simbar M, Bidhendi Yarandi R, Minooee S, Hollis BW, Hosseinpanah F.J Clin Endocrinol Metab. 2018 Aug 1;103(8):2936-2948. doi: 10.1210/jc.2018-00109.
It would be useful to add a paragraph about use of calcifediol in treating COVID-19 in Barcelona and, perhaps, other cities in Spain. The advantage of it over cholecalciferol is the rapidity with which it increases 25(OH)D concentration. That is important since two of the important mechanisms whereby vitamin D reduces severity of COVID-19 is by inducing production of cathelicidin, which can kill the virus, and regulating cytokine production towards the anti-inflammatory state, thereby reducing risk of the cytokine storm that can be deadly.
“In obesity and malabsorptive conditions, calcifediol emerges as a more effective alternative to cholecalciferol due to its superior bioavailability and resistance to sequestration in adipose tissue.”
Comment: I do not think this statement has been well supported in the text.
I suggest searching Google Scholar with “obesity, supplement with cholecaciferol or calcifediol”
Calcifediol is also much more expensive in many countries although probably not in Spain where both types are by prescription. Also note that one of the reasons why obese people have lower 25(OH)D concentrations is volumetric dilution. Also, their diet may have more carbohydrates and less animal products with vitamin D including both meat and fish.
Plasma concentrations of 25-hydroxyvitamin D in meat eaters, fish eaters, vegetarians and vegans: results from the EPIC-Oxford study.
Crowe FL, Steur M, Allen NE, Appleby PN, Travis RC, Key TJ.Public Health Nutr. 2011 Feb;14(2):340-6. doi: 10.1017/S1368980010002454.
Natural Vitamin D in Food: To What Degree Does 25-Hydroxyvitamin D Contribute to the Vitamin D Activity in Food?
Jakobsen J, Christensen T.JBMR Plus. 2021 Jan 3;5(1):e10453. doi: 10.1002/jbm4.10453.
Author Response
Furthermore, vitamin D supplementation not only helps prevent bone fractures, but may also slow the progression of various conditions, such as diabetes, autoimmune disorders, cardiovascular disease, and cancer.
Comment: Anything mentioned in the abstract should also be discussed in the text. Cancer is not discussed in the text. I suggest searching Google Scholar for one or two reviews on vitamin D and cancer. Also, one or two reviews regarding cardiovascular disease, diabetes (type 2 diabetes mellitus), and autoimmune diseases. Thank you. We decided to remove cancer from the topics of the paper, and from the abstract because it isn’t the focus of the paper.
From there, 25(OH)D is further converted in the kidneys to its biologically active form, 1,25-dihydroxyvitamin D [1,25(OH)2D], which exerts its effects through the vitamin D receptor expressed in various tissues [2,3]
Comment: Please discuss extra-renal conversion to 1,25D. Zehnder and Hewison published an early paper on this topic. It is very important for cancer since the organs with cancer will do it for protection. Please, see above.
One of the critical functions of vitamin D is to enhance intestinal calcium absorption,
Comment: Also phosphorus absorption. Thank you, we added it in the text.
Cholecalciferol
Half-Life in Blood |
Long (weeks to months) |
Comment: I think it is a day or two. What is the reference for this statement.
Calcifediol
Intermediate (2–3 weeks) |
Comment; It is stored in the muscles and put back in the blood when needed, such as in winter:
Mason RS, Rybchyn MS, Abboud M, Brennan-Speranza TC, Fraser DR.The Role of Skeletal Muscle in Maintaining Vitamin D Status in Winter. Curr Dev Nutr. 2019 Jul 25;3(10):nzz087. doi: 10.1093/cdn/nzz087.
Rybchyn MS, Abboud M, Puglisi DA, Gordon-Thomson C, Brennan-Speranza TC, Mason RS, Fraser DR Skeletal Muscle and the Maintenance of Vitamin D Status. Nutrients. 2020 Oct 26;12(11):3270. doi: 10.3390/nu12113270.
Thank you for your comment. We decide to eliminate the “half-life in blood” because it could have been confusing. We replaced it with “duration of action”.
Thanks to a better intestinal absorption and a greater hydrophilic feature, calcifediol results in a faster and more predictable increase in vitamin D levels,
Comment: If vitamin D levels were written as 25(OH)D concentration, it would be readily apparent that calcifediol is 25(OH)D, so does not go through the D to 25(OH)D conversion step.
Since “plasma 25(OH)D concentrations” is used a few lines later, please use that instead of vitamin D levels.
Thank you for clarification. We replaced it in the text.
According to Barbara J Boucher, the primary challenge in evaluating randomized controlled trials about vitamin D supplementation is considering vitamin D a drug rather than a nutrient [15].
Comment: This statement should be expanded. The problem is that in most vitamin D RCTs, participants are enrolled with high 25(OH)D concentrations, are given low vitamin D doses, and the outcomes are based on intention to treat, not achieved 25(OH)D concentration.
A couple additional references to consider:
Critical Appraisal of Large Vitamin D Randomized Controlled Trials.
Pilz S, Trummer C, Theiler-Schwetz V, Grübler MR, Verheyen ND, Odler B, Karras SN, Zittermann A, März W. Nutrients. 2022 Jan 12;14(2):303. doi: 10.3390/nu14020303.
Intervention Approaches in Studying the Response to Vitamin D3 Supplementation.
Gospodarska E, Ghosh Dastidar R, Carlberg C. Nutrients. 2023 Jul 29;15(15):3382. doi: 10.3390/nu15153382.
Thank you for pointing this out. We inserted the comment and the reference you mentioned, following the sentence regarding Barbara J Boucher in the chapter “vitamin D and obesity”
Prior to randomization, an inverse relationship between serum total 25(OH)D levels and BMI was observed [16]. After two years, increases in 25(OH)D levels were significantly lower with higher BMI categories [16].
Comment: According to Figure S1(B) in [16], increases in 25(OH)D were very similar for all three BMI groups.
Thank you for your comment, we corrected it in the text.
50] C. Palacios, LL. Kostiuk, A. Cuthbert, and J. Weeks, ‘Vitamin D supplementation for women during pregnancy’, Cochrane Database of Systematic Reviews, 2024, doi: 10.1002/14651858.CD008873.pub5
Comment: As this review was not very supportive of vitamin D for pregnant women, I recommend citing this article as well:
Effectiveness of Prenatal Vitamin D Deficiency Screening and Treatment Program: A Stratified Randomized Field Trial.
Rostami M, Tehrani FR, Simbar M, Bidhendi Yarandi R, Minooee S, Hollis BW, Hosseinpanah F.J Clin Endocrinol Metab. 2018 Aug 1;103(8):2936-2948. doi: 10.1210/jc.2018-00109.
Thank you for this precious citation. We replaced it.
It would be useful to add a paragraph about use of calcifediol in treating COVID-19 in Barcelona and, perhaps, other cities in Spain. The advantage of it over cholecalciferol is the rapidity with which it increases 25(OH)D concentration. That is important since two of the important mechanisms whereby vitamin D reduces severity of COVID-19 is by inducing production of cathelicidin, which can kill the virus, and regulating cytokine production towards the anti-inflammatory state, thereby reducing risk of the cytokine storm that can be deadly. Thank you for your proposal. However, we think that vit D potential effects on immune response is out of the main scope of our paper. We could use this reference in another more relevant article.
“In obesity and malabsorptive conditions, calcifediol emerges as a more effective alternative to cholecalciferol due to its superior bioavailability and resistance to sequestration in adipose tissue.”
Comment: I do not think this statement has been well supported in the text.
I suggest searching Google Scholar with “obesity, supplement with cholecaciferol or calcifediol”
Calcifediol is also much more expensive in many countries although probably not in Spain where both types are by prescription. Also note that one of the reasons why obese people have lower 25(OH)D concentrations is volumetric dilution. Also, their diet may have more carbohydrates and less animal products with vitamin D including both meat and fish.
Plasma concentrations of 25-hydroxyvitamin D in meat eaters, fish eaters, vegetarians and vegans: results from the EPIC-Oxford study.
Crowe FL, Steur M, Allen NE, Appleby PN, Travis RC, Key TJ.Public Health Nutr. 2011 Feb;14(2):340-6. doi: 10.1017/S1368980010002454.
Natural Vitamin D in Food: To What Degree Does 25-Hydroxyvitamin D Contribute to the Vitamin D Activity in Food?
Jakobsen J, Christensen T.JBMR Plus. 2021 Jan 3;5(1):e10453. doi: 10.1002/jbm4.10453.
Thank you for your comment. We have explained the concept better in the text, citing the suggested articles.

Round 2
Reviewer 4 Report
Comments and Suggestions for Authors
The manuscript is much improved.
No further suggestions.